# Training Very Deep Networks

**Rupesh Kumar Srivastava    Klaus Greff    Jürgen Schmidhuber**

The Swiss AI Lab IDSIA / USI / SUPSI
`{rupesh, klaus, juergen}@idsia.ch`

## Abstract

Theoretical and empirical evidence indicates that the depth of neural networks is crucial for their success. However, training becomes more difficult as depth increases, and training of very deep networks remains an open problem. Here we introduce a new architecture designed to overcome this. Our so-called highway networks allow unimpeded information flow across many layers on *information highways*. They are inspired by Long Short-Term Memory recurrent networks and use adaptive gating units to regulate the information flow. Even with hundreds of layers, highway networks can be trained directly through simple gradient descent. This enables the study of extremely deep and efficient architectures.

## 1  Introduction & Previous Work

Many recent empirical breakthroughs in supervised machine learning have been achieved through large and deep neural networks. Network depth (the number of successive computational layers) has played perhaps the most important role in these successes. For instance, within just a few years, the top-5 image classification accuracy on the 1000-class ImageNet dataset has increased from $\sim$84% [1] to $\sim$95% [2, 3] using deeper networks with rather small receptive fields [4, 5]. Other results on practical machine learning problems have also underscored the superiority of deeper networks [6] in terms of accuracy and/or performance.

In fact, deep networks can represent certain function classes far more efficiently than shallow ones. This is perhaps most obvious for recurrent nets, the deepest of them all. For example, the $n$ bit parity problem can in principle be learned by a large feedforward net with $n$ binary input units, 1 output unit, and a single but large hidden layer. But the natural solution for arbitrary $n$ is a recurrent net with only 3 units and 5 weights, reading the input bit string one bit at a time, making a single recurrent hidden unit flip its state whenever a new 1 is observed [7]. Related observations hold for Boolean circuits [8, 9] and modern neural networks [10, 11, 12].

To deal with the difficulties of training deep networks, some researchers have focused on developing better optimizers (e.g. [13, 14, 15]). Well-designed initialization strategies, in particular the normalized variance-preserving initialization for certain activation functions [16, 17], have been widely adopted for training moderately deep networks. Other similarly motivated strategies have shown promising results in preliminary experiments [18, 19]. Experiments showed that certain activation functions based on local competition [20, 21] may help to train deeper networks. Skip connections between layers or to output layers (where error is "injected") have long been used in neural networks, more recently with the explicit aim to improve the flow of information [22, 23, 2, 24]. A related recent technique is based on using soft targets from a shallow teacher network to aid in training deeper student networks in multiple stages [25], similar to the neural history compressor for sequences, where a slowly ticking teacher recurrent net is "distilled" into a quickly ticking student recurrent net by forcing the latter to predict the hidden units of the former [26]. Finally, deep networks can be trained layer-wise to help in credit assignment [26, 27], but this approach is less attractive compared to direct training.

Very deep network training still faces problems, albeit perhaps less fundamental ones than the problem of vanishing gradients in standard recurrent networks [28]. The stacking of several non-linear transformations in conventional feed-forward network architectures typically results in poor propagation of activations and gradients. Hence it remains hard to investigate the benefits of very deep networks for a variety of problems.

To overcome this, we take inspiration from Long Short Term Memory (LSTM) recurrent networks [29, 30]. We propose to modify the architecture of very deep feedforward networks such that information flow across layers becomes much easier. This is accomplished through an LSTM-inspired adaptive gating mechanism that allows for computation paths along which information can flow across many layers without attenuation. We call such paths *information highways*. They yield *highway networks*, as opposed to traditional 'plain' networks.[1]

Our primary contribution is to show that extremely deep highway networks can be trained directly using stochastic gradient descent (SGD), in contrast to plain networks which become hard to optimize as depth increases (Section 3.1). Deep networks with limited computational budget (for which a two-stage training procedure mentioned above was recently proposed [25]) can also be directly trained in a single stage when converted to highway networks. Their ease of training is supported by experimental results demonstrating that highway networks also generalize well to unseen data.

## 2 Highway Networks

**Notation**  We use boldface letters for vectors and matrices, and italicized capital letters to denote transformation functions. $\mathbf{0}$ and $\mathbf{1}$ denote vectors of zeros and ones respectively, and $\mathbf{I}$ denotes an identity matrix. The function $\sigma(x)$ is defined as $\sigma(x) = \frac{1}{1+e^{-x}}, x \in \mathbb{R}$. The dot operator $(\cdot)$ is used to denote element-wise multiplication.

A *plain* feedforward neural network typically consists of $L$ layers where the $l^{th}$ layer ($l \in \{1, 2, ..., L\}$) applies a non-linear transformation $H$ (parameterized by $\mathbf{W_{H,l}}$) on its input $\mathbf{x_l}$ to produce its output $\mathbf{y_l}$. Thus, $\mathbf{x_1}$ is the input to the network and $\mathbf{y_L}$ is the network's output. Omitting the layer index and biases for clarity,

$$\mathbf{y} = H(\mathbf{x}, \mathbf{W_H}). \tag{1}$$

$H$ is usually an affine transform followed by a non-linear activation function, but in general it may take other forms, possibly convolutional or recurrent. For a highway network, we additionally define two non-linear transforms $T(\mathbf{x}, \mathbf{W_T})$ and $C(\mathbf{x}, \mathbf{W_C})$ such that

$$\mathbf{y} = H(\mathbf{x}, \mathbf{W_H}) \cdot T(\mathbf{x}, \mathbf{W_T}) + \mathbf{x} \cdot C(\mathbf{x}, \mathbf{W_C}). \tag{2}$$

We refer to $T$ as the *transform* gate and $C$ as the *carry* gate, since they express how much of the output is produced by transforming the input and carrying it, respectively. For simplicity, in this paper we set $C = 1 - T$, giving

$$\mathbf{y} = H(\mathbf{x}, \mathbf{W_H}) \cdot T(\mathbf{x}, \mathbf{W_T}) + \mathbf{x} \cdot (1 - T(\mathbf{x}, \mathbf{W_T})). \tag{3}$$

The dimensionality of $\mathbf{x}, \mathbf{y}, H(\mathbf{x}, \mathbf{W_H})$ and $T(\mathbf{x}, \mathbf{W_T})$ must be the same for Equation 3 to be valid. Note that this layer transformation is much more flexible than Equation 1. In particular, observe that for particular values of $T$,

$$\mathbf{y} = \begin{cases} \mathbf{x}, & \text{if } T(\mathbf{x}, \mathbf{W_T}) = \mathbf{0}, \\ H(\mathbf{x}, \mathbf{W_H}), & \text{if } T(\mathbf{x}, \mathbf{W_T}) = \mathbf{1}. \end{cases} \tag{4}$$

Similarly, for the Jacobian of the layer transform,

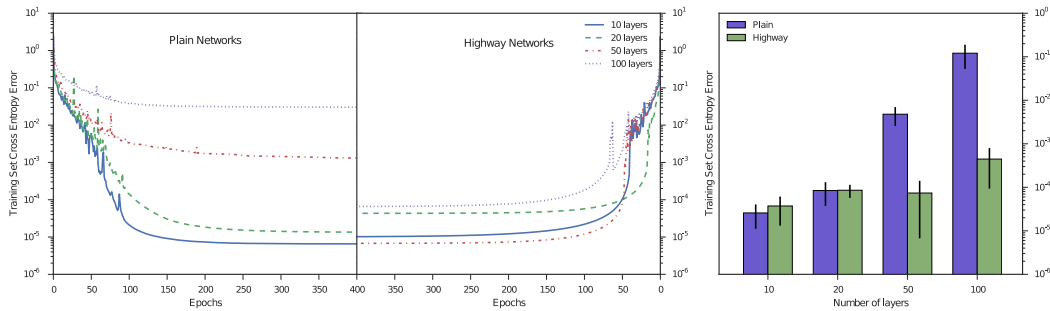

Figure 1: Comparison of optimization of plain networks and highway networks of various depths. *Left:* The training curves for the best hyperparameter settings obtained for each network depth. *Right:* Mean performance of top 10 (out of 100) hyperparameter settings. Plain networks become much harder to optimize with increasing depth, while highway networks with up to 100 layers can still be optimized well. Best viewed on screen (larger version included in Supplementary Material).

$$\frac{d\mathbf{y}}{d\mathbf{x}} = \begin{cases} \mathbf{I}, & \text{if } T(\mathbf{x}, \mathbf{W_T}) = \mathbf{0}, \\ H'(\mathbf{x}, \mathbf{W_H}), & \text{if } T(\mathbf{x}, \mathbf{W_T}) = \mathbf{1}. \end{cases} \tag{5}$$

Thus, depending on the output of the transform gates, a highway layer can smoothly vary its behavior between that of $H$ and that of a layer which simply passes its inputs through. Just as a plain layer consists of multiple computing units such that the $i^{th}$ unit computes $y_i = H_i(\mathbf{x})$, a highway network consists of multiple **blocks** such that the $i^{th}$ block computes a **block state** $H_i(\mathbf{x})$ and **transform gate output** $T_i(\mathbf{x})$. Finally, it produces the **block output** $y_i = H_i(\mathbf{x}) * T_i(\mathbf{x}) + x_i * (1 - T_i(\mathbf{x}))$, which is connected to the next layer.[2]

## 2.1 Constructing Highway Networks

As mentioned earlier, Equation 3 requires that the dimensionality of $\mathbf{x}, \mathbf{y}, H(\mathbf{x}, \mathbf{W_H})$ and $T(\mathbf{x}, \mathbf{W_T})$ be the same. To change the size of the intermediate representation, one can replace $\mathbf{x}$ with $\hat{\mathbf{x}}$ obtained by suitably sub-sampling or zero-padding $\mathbf{x}$. Another alternative is to use a plain layer (without highways) to change dimensionality, which is the strategy we use in this study.

Convolutional highway layers utilize weight-sharing and local receptive fields for both $H$ and $T$ transforms. We used the same sized receptive fields for both, and zero-padding to ensure that the block state and transform gate feature maps match the input size.

## 2.2 Training Deep Highway Networks

We use the transform gate defined as $T(\mathbf{x}) = \sigma(\mathbf{W_T}^T\mathbf{x} + \mathbf{b_T})$, where $\mathbf{W_T}$ is the weight matrix and $\mathbf{b_T}$ the bias vector for the transform gates. This suggests a simple initialization scheme which is independent of the nature of $H$: $b_T$ can be initialized with a negative value (e.g. -1, -3 etc.) such that the network is initially biased towards *carry* behavior. This scheme is strongly inspired by the proposal [30] to initially bias the gates in an LSTM network, to help bridge long-term temporal dependencies early in learning. Note that $\sigma(x) \in (0, 1), \forall x \in \mathbb{R}$, so the conditions in Equation 4 can never be met exactly.

In our experiments, we found that a negative bias initialization for the transform gates was sufficient for training to proceed in very deep networks for various zero-mean initial distributions of $W_H$ and different activation functions used by $H$. In pilot experiments, SGD did not stall for networks with more than 1000 layers. Although the initial bias is best treated as a hyperparameter, as a general guideline we suggest values of -1, -2 and -3 for convolutional highway networks of depth approximately 10, 20 and 30.

| Network | Highway Networks | | Maxout [20] | DSN [24] |
|---|---|---|---|---|
| | 10-layer (width 16) | 10-layer (width 32) | | |
| No. of parameters | 39 K | 151 K | 420 K | 350 K |
| Test Accuracy (in %) | 99.43 (99.4±0.03) | 99.55 (99.54±0.02) | 99.55 | 99.61 |

Table 1: Test set classification accuracy for pilot experiments on the MNIST dataset.

| Network | No. of Layers | No. of Parameters | Accuracy (in %) |
|---|---|---|---|
| Fitnet Results (reported by Romero et. al.[25]) | | | |
| Teacher | 5 | ∼9M | 90.18 |
| Fitnet A | 11 | ∼250K | 89.01 |
| Fitnet B | 19 | ∼2.5M | 91.61 |
| Highway networks | | | |
| Highway A (Fitnet A) | 11 | ∼236K | 89.18 |
| Highway B (Fitnet B) | 19 | ∼2.3M | **92.46 (92.28±0.16)** |
| Highway C | 32 | ∼1.25M | 91.20 |

Table 2: CIFAR-10 test set accuracy of convolutional highway networks. Architectures tested were based on *fitnets* trained by Romero et. al. [25] using two-stage hint based training. Highway networks were trained in a single stage without hints, matching or exceeding the performance of fitnets.

# 3 Experiments

All networks were trained using SGD with momentum. An exponentially decaying learning rate was used in Section 3.1. For the rest of the experiments, a simpler commonly used strategy was employed where the learning rate starts at a value $\lambda$ and decays according to a fixed schedule by a factor $\gamma$. $\lambda$, $\gamma$ and the schedule were selected once based on validation set performance on the CIFAR-10 dataset, and kept fixed for all experiments. All convolutional highway networks utilize the rectified linear activation function [16] to compute the block state $H$. To provide a better estimate of the variability of classification results due to random initialization, we report our results in the format *Best (mean ± std.dev.)* based on 5 runs wherever available. Experiments were conducted using Caffe [33] and Brainstorm (https://github.com/IDSIA/brainstorm) frameworks. Source code, hyperparameter search results and related scripts are publicly available at http://people.idsia.ch/~rupesh/very_deep_learning/.

## 3.1 Optimization

To support the hypothesis that highway networks do not suffer from increasing depth, we conducted a series of rigorous optimization experiments, comparing them to plain networks with normalized initialization [16, 17].

We trained both plain and highway networks of varying varying depths on the MNIST digit classification dataset. All networks are *thin*: each layer has 50 blocks for highway networks and 71 units for plain networks, yielding roughly identical numbers of parameters ($\approx$5000) per layer. In all networks, the first layer is a fully connected plain layer followed by 9, 19, 49, or 99 fully connected plain or highway layers. Finally, the network output is produced by a *softmax* layer. We performed a random search of 100 runs for both plain and highway networks to find good settings for the following hyperparameters: initial learning rate, momentum, learning rate exponential decay factor & activation function (either *rectified linear* or *tanh*). For highway networks, an additional hyperparameter was the initial value for the transform gate bias (between -1 and -10). Other weights were initialized using the same normalized initialization as plain networks.

The training curves for the best performing networks for each depth are shown in Figure 1. As expected, 10 and 20-layer plain networks exhibit very good performance (mean loss $< 1e^{-4}$), which significantly degrades as depth increases, even though network capacity increases. Highway networks do not suffer from an increase in depth, and 50/100 layer highway networks perform similar to 10/20 layer networks. The 100-layer highway network performed more than 2 orders of magnitude better compared to a similarly-sized plain network. It was also observed that highway networks consistently converged significantly faster than plain ones.

| Network | CIFAR-10 Accuracy (in %) | CIFAR-100 Accuracy (in %) |
|---|---|---|
| Maxout [20] | 90.62 | 61.42 |
| dasNet [36] | 90.78 | 66.22 |
| NiN [35] | 91.19 | 64.32 |
| DSN [24] | 92.03 | 65.43 |
| All-CNN [37] | **92.75** | 66.29 |
| Highway Network | 92.40 (92.31±0.12) | **67.76 (67.61±0.15)** |

Table 3: Test set accuracy of convolutional highway networks on the CIFAR-10 and CIFAR-100 object recognition datasets with typical data augmentation. For comparison, we list the accuracy reported by recent studies in similar experimental settings.

## 3.2 Pilot Experiments on MNIST Digit Classification

As a sanity check for the generalization capability of highway networks, we trained 10-layer convolutional highway networks on MNIST, using two architectures, each with 9 convolutional layers followed by a softmax output. The number of filter maps (width) was set to 16 and 32 for all the layers. We obtained test set performance competitive with state-of-the-art methods with much fewer parameters, as show in Table 1.

## 3.3 Experiments on CIFAR-10 and CIFAR-100 Object Recognition

### 3.3.1 Comparison to Fitnets

**Fitnet training** Maxout networks can cope much better with increased depth than those with traditional activation functions [20]. However, Romero et. al. [25] recently reported that training on CIFAR-10 through plain backpropogation was only possible for maxout networks with a depth up to 5 layers when the number of parameters was limited to ∼250K and the number of multiplications to ∼30M. Similar limitations were observed for higher computational budgets. Training of deeper networks was only possible through the use of a two-stage training procedure and addition of soft targets produced from a pre-trained shallow teacher network (hint-based training).

We found that it was easy to train highway networks with numbers of parameters and operations comparable to those of fitnets in a single stage using SGD. As shown in Table 2, Highway A and Highway B, which are based on the architectures of Fitnet A and Fitnet B, respectively, obtain similar or higher accuracy on the test set. We were also able to train thinner and deeper networks: for example a 32-layer highway network consisting of alternating receptive fields of size 3x3 and 1x1 with ∼1.25M parameters performs better than the earlier teacher network [20].

### 3.3.2 Comparison to State-of-the-art Methods

It is possible to obtain high performance on the CIFAR-10 and CIFAR-100 datasets by utilizing very large networks and extensive data augmentation. This approach was popularized by Ciresan et. al. [5] and recently extended by Graham [34]. Since our aim is only to demonstrate that deeper networks can be trained without sacrificing ease of training or generalization ability, we only performed experiments in the more common setting of global contrast normalization, small translations and mirroring of images. Following Lin et. al. [35], we replaced the fully connected layer used in the networks in the previous section with a convolutional layer with a receptive field of size one and a global average pooling layer. The hyperparameters from the last section were re-used for both CIFAR-10 and CIFAR-100, therefore it is quite possible to obtain much better results with better architectures/hyperparameters. The results are tabulated in Table 3.

## 4  Analysis

Figure 2 illustrates the inner workings of the best[3] 50 hidden layer fully-connected highway networks trained on MNIST (top row) and CIFAR-100 (bottom row). The first three columns show

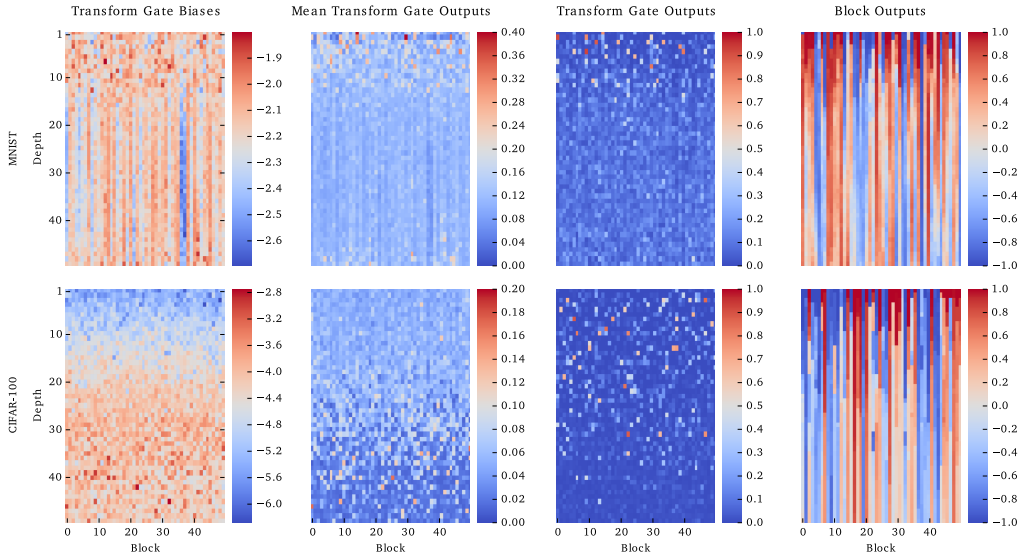

Figure 2: Visualization of best 50 hidden-layer highway networks trained on MNIST (top row) and CIFAR-100 (bottom row). The first hidden layer is a plain layer which changes the dimensionality of the representation to 50. Each of the 49 highway layers (y-axis) consists of 50 blocks (x-axis). The first column shows the transform gate biases, which were initialized to -2 and -4 respectively. In the second column the mean output of the transform gate over all training examples is depicted. The third and fourth columns show the output of the transform gates and the block outputs (both networks using tanh) for a single random training sample. Best viewed in color.

the bias, the mean activity over all training samples, and the activity for a single random sample for each transform gate respectively. Block outputs for the same single sample are displayed in the last column.

The transform gate biases of the two networks were initialized to -2 and -4 respectively. It is interesting to note that contrary to our expectations most biases decreased further during training. For the CIFAR-100 network the biases increase with depth forming a gradient. Curiously this gradient is inversely correlated with the average activity of the transform gates, as seen in the second column. This indicates that the strong negative biases at low depths are not used to shut down the gates, but to make them more selective. This behavior is also suggested by the fact that the transform gate activity for a single example (column 3) is very sparse. The effect is more pronounced for the CIFAR-100 network, but can also be observed to a lesser extent in the MNIST network.

The last column of Figure 2 displays the block outputs and visualizes the concept of "information highways". Most of the outputs stay constant over many layers forming a pattern of stripes. Most of the change in outputs happens in the early layers ($\approx 15$ for MNIST and $\approx 40$ for CIFAR-100).

## 4.1 Routing of Information

One possible advantage of the highway architecture over hard-wired shortcut connections is that the network can learn to dynamically adjust the routing of the information based on the current input. This begs the question: does this behaviour manifest itself in trained networks or do they just learn a static routing that applies to all inputs similarly. A partial answer can be found by looking at the mean transform gate activity (second column) and the single example transform gate outputs (third column) in Figure 2. Especially for the CIFAR-100 case, most transform gates are active on average, while they show very selective activity for the single example. This implies that for each sample only a few blocks perform transformation but different blocks are utilized by different samples.

This data-dependent routing mechanism is further investigated in Figure 3. In each of the columns we show how the average over all samples of one specific class differs from the total average shown in the second column of Figure 2. For MNIST digits 0 and 7 substantial differences can be seen

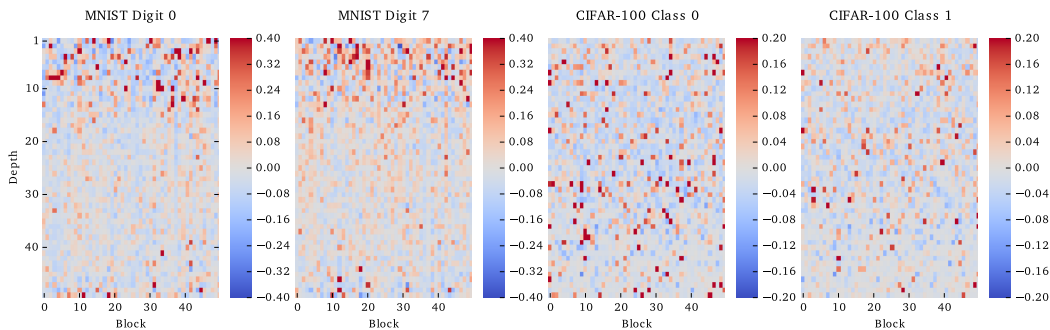

Figure 3: Visualization showing the extent to which the mean transform gate activity for certain classes differs from the mean activity over all training samples. Generated using the same 50-layer highway networks on MNIST on CIFAR-100 as Figure 2. Best viewed in color.

within the first 15 layers, while for CIFAR class numbers 0 and 1 the differences are sparser and spread out over all layers. In both cases it is clear that the mean activity pattern differs between classes. The gating system acts not just as a mechanism to ease training, but also as an important part of the computation in a trained network.

## 4.2 Layer Importance

Since we bias all the transform gates towards being closed, in the beginning every layer mostly copies the activations of the previous layer. Does training indeed change this behaviour, or is the final network still essentially equivalent to a network with a much fewer layers? To shed light on this issue, we investigated the extent to which *lesioning* a single layer affects the total performance of trained networks from Section 3.1. By lesioning, we mean manually setting all the transform gates of a layer to 0 forcing it to simply copy its inputs. For each layer, we evaluated the network on the full training set with the gates of that layer closed. The resulting performance as a function of the lesioned layer is shown in Figure 4.

For MNIST (left) it can be seen that the error rises significantly if any one of the early layers is removed, but layers $15 - 45$ seem to have close to no effect on the final performance. About 60% of the layers don't learn to contribute to the final result, likely because MNIST is a simple dataset that doesn't require much depth.

We see a different picture for the CIFAR-100 dataset (right) with performance degrading noticeably when removing any of the first $\approx 40$ layers. This suggests that for complex problems a highway network can learn to utilize all of its layers, while for simpler problems like MNIST it will keep many of the unneeded layers idle. Such behavior is desirable for deep networks in general, but appears difficult to obtain using plain networks.

## 5 Discussion

Alternative approaches to counter the difficulties posed by depth mentioned in Section 1 often have several limitations. Learning to route information through neural networks with the help of competitive interactions has helped to scale up their application to challenging problems by improving credit assignment [38], but they still suffer when depth increases beyond $\approx$20 even with careful initialization [17]. Effective initialization methods can be difficult to derive for a variety of activation functions. Deep supervision [24] has been shown to hurt performance of thin deep networks [25].

Very deep highway networks, on the other hand, can directly be trained with simple gradient descent methods due to their specific architecture. This property does not rely on specific non-linear transformations, which may be complex convolutional or recurrent transforms, and derivation of a suitable initialization scheme is not essential. The additional parameters required by the gating mechanism help in routing information through the use of multiplicative connections, responding differently to different inputs, unlike fixed "skip" connections.

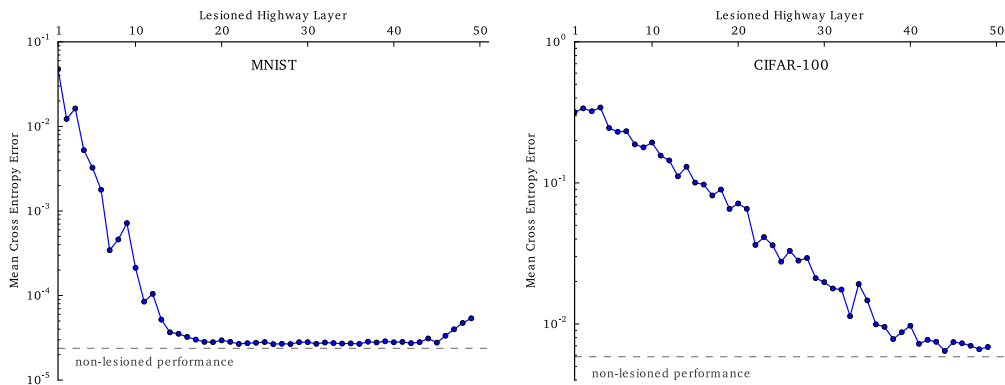

Figure 4: Lesioned training set performance (y-axis) of the best 50-layer highway networks on MNIST (left) and CIFAR-100 (right), as a function of the lesioned layer (x-axis). Evaluated on the full training set while forcefully closing all the transform gates of a single layer at a time. The non-lesioned performance is indicated as a dashed line at the bottom.

A possible objection is that many layers might remain unused if the transform gates stay closed. Our experiments show that this possibility does not affect networks adversely—deep and narrow highway networks can match/exceed the accuracy of wide and shallow maxout networks, which would not be possible if layers did not perform useful computations. Additionally, we can exploit the structure of highways to directly evaluate the contribution of each layer as shown in Figure 4. For the first time, highway networks allow us to examine how much computation depth is needed for a given problem, which can not be easily done with plain networks.

### Acknowledgments

We thank NVIDIA Corporation for their donation of GPUs and acknowledge funding from the EU project NASCENCE (FP7-ICT-317662). We are grateful to Sepp Hochreiter and Thomas Unterthiner for helpful comments and Jan Koutník for help in conducting experiments.

## Footnotes

[1]This paper expands upon a shorter report on Highway Networks [31]. More recently, a similar LSTM-inspired model was also proposed [32].

[2]Our pilot experiments on training very deep networks were successful with a more complex block design closely resembling an LSTM block "unrolled in time". Here we report results only for a much simplified form.

[3]obtained via random search over hyperparameters to minimize the best training set error achieved using each configuration

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
