[Supplementary Material]

# Supplementary Material:
# Training Very Deep Networks

**Rupesh Kumar Srivastava**    **Klaus Greff**    **Jürgen Schmidhuber**

The Swiss AI Lab IDSIA
{rupesh, klaus, juergen}@idsia.ch

## 1  Highway Networks Implementation

**Notation**   We use boldface letters for vectors and matrices, and italicized capital letters to denote transformation functions. $\mathbf{0}$ and $\mathbf{1}$ denote vectors of zeros and ones respectively, and $\mathbf{I}$ denotes an identity matrix. The dot operator ($\cdot$) is used to denote element-wise multiplication.

In most modular and efficient implementations, neural networks are represented as a series of simpler *operations* chained together. Let's assume that some non-linear transformations $H$, $T$ and $C$ are already defined so that for input $\mathbf{x}$ and some transformation parameters (to be learned) $\mathbf{W_H}, \mathbf{W_T}, \mathbf{W_C}$:

$$
\begin{aligned}
\mathbf{H} &= H(\mathbf{x}, \mathbf{W_H}), \\
\mathbf{T} &= T(\mathbf{x}, \mathbf{W_T}), \\
\mathbf{C} &= C(\mathbf{x}, \mathbf{W_C}).
\end{aligned}
\tag{1}
$$

Typically $H$ would be an affine transformation followed by a non-linear activation function such as *tanh* or *rectified linear* for a feedforward network, but in general it may take convolutional, recurrent or other forms. Similarly, $T$ and $C$ can take any form but should typically map inputs to values in (0, 1), since they are interpreted as gates.

We define a **Highway** operation simply in terms of $\mathbf{x}, \mathbf{T}, \mathbf{H}$ and $\mathbf{C}$:

$$
\mathbf{y} = \mathbf{H} \cdot \mathbf{T} + \mathbf{x} \cdot \mathbf{C},
\tag{2}
$$

which essentially implements what's usually called the *forward pass* of the operation using element-wise multiplication and addition operations.

In this paper, we have set $\mathbf{C} = 1 - \mathbf{T}$ for simplicity, giving

$$
\mathbf{y} = \mathbf{H} \cdot \mathbf{T} + \mathbf{x} \cdot (\mathbf{1} - \mathbf{T}).
\tag{3}
$$

*Backward pass*: The **Highway** operation utilizes no additional parameters on its own, so during backpropagation, only the derivatives of $\mathbf{x}, \mathbf{T}, \mathbf{H}$ need to be computed. These are simply:

$$
\begin{aligned}
\mathbf{dH} &= \mathbf{T} \cdot \mathbf{dy}, \\
\mathbf{dT} &= (\mathbf{H} - \mathbf{x}) \cdot \mathbf{dy}, \\
\mathbf{dx} &= (\mathbf{1} - \mathbf{T}) \cdot \mathbf{dy}.
\end{aligned}
\tag{4}
$$

Figure 1: Comparison of optimization of plain networks and highway networks of various depths. All networks were optimized using SGD with momentum. *Left*: The training curves for the best hyperparameter settings obtained for each network depth. *Right*: Mean performance of top 10 (out of 100) hyperparameter settings. Plain networks become much harder to optimize with increasing depth, while highway networks with up to 100 layers can still be optimized well.