[Reviews · NeurIPS 2015]

Submitted by Assigned_Reviewer_1

I like the paper, the idea is well described and the experiments are convincing to a certain degree. The best thing in my opinion is that the authors tried to analyze the learned networks with respect to the pattern of gate outputs.

(1) The gate unit T is modeled with a standard sigmoid activation. Therefore, the response value is never exactly 0 or 1 (this was also stated by the authors) and the gradients in eq. 5 are not correct. The authors should explain in the paper how backpropagation is exactly performed in these networks.

(2) The effect of the initialization is always tricky to analyze, however, the fact that the authors propose to use constant negative biases depending on the number of levels is also not satisfying. I would like to see a plot of the performance with respect to the initial value of the bias. The initial bias acts as a kind of regularizer in this case with a high negative bias supporting networks with "longer highways" (I like this comparison by the way). (3) "report by recent studies in similar experimental settings" This sentence is unsatisfying. Either the methods use the same experimental setting, which would allow for comparison, or not, which would forbid a direct comparison. The authors need to clarify the differences of the experimental setups in the paper. (4) Giving "max" accuracies in tables is rather uncommon and also not reasonable from a statistical point of view. The authors should therefore always give mean accuracy and standard deviation for every method (and when available). (5) It would be interesting to study the proposed approach from a learning theory perspective.
Summary: The paper describes how to use additional gate units in neural networks that allow for layers to simply act as an identity. The gate unit response depends on the input and therefore allows propagating inputs through a large number of very selective layers.

Submitted by Assigned_Reviewer_2

This paper proposes an architecture for very deep networks which enables input to be "carried" unchanged through a layer using gates dependent on the input itself.

This enables much deeper networks to be trained than would be possible with a standard feed forward architecture.

The authors perform a very interesting analysis of the trained networks to determine the manner in which the gates select their inputs in in different layers of the network, which is exactly what I would have wanted to see.

The paper is of high quality, clearly written, and original.

Though there is naturally some similarity to other recent work (notably FitNets, and LSTMs), the differences are clearly explained and where appropriate, they are compared.

Promising results are achieved.

I have no suggested improvements.

Note "Tables 3 and 3" (p5).

Summary: This is a very solid paper.

The proposed technique is novel and achieves good results.

The analysis is interesting and thorough.

Submitted by Assigned_Reviewer_3

This paper proposes the use of LSTM units as a method of training very deep networks.

This is a high-quality, well-written paper. The idea is original, and the results are quite interesting.

Machine learning researchers frequently find that they get better results by adding more and more layers to their neural networks, but the difficulties of initialization and decaying/exploding gradients have been severely limiting. Indeed, the difficulties of getting information to flow through deep neural networks arguably kept them out of widespread use for 30 years. This paper addresses this problem head on and demonstrates one method for training 100 layer nets.

In particular, the experiments showing the routing of information and layer importance are very cool.

Specific comments:

It seemed that in Section 4.1, 'most transform blocks are active on average' should be changed to 'not active on average.'

One could initialize each layer of a very deep network with the identity matrix (possibly with a small amount of noise). It might be worth commenting how transform gates differentiate highway networks from this simple scheme.
Summary: This is a great paper, demonstrating a method for training very deep neural networks, and was a pleasure to read.

Submitted by Assigned_Reviewer_4

The paper proposes a generalization of training neural networks which have many stacked layers. The information highway idea works well illustrated by experimentation. Further, authors have shown classification accuracy results can be improved by this method.

Quality: the idea seems novel, and experimentation is solid. Analysis sections could use a bit more variety of details or illustrations.

Clarity: the main messages of the paper are well delivered and quite clear.

Originality: the proposed method is sufficiently original although a generalization of previous neural network techniques.

Significant: the paper illustrates significance to the field since deeper and deeper neural networks need to be trained to arrive at better results in application areas.

Summary: The paper describes an affective method to train very deep neural networks by means of 'information highways', or building direct connections to upper network layers. Although a generalization of prior techniques, such as cross-layer connections, the authors have shown this method to be effective by experimentation. The contributions are quite novel and well supported by experimental evidence.

Submitted by Assigned_Reviewer_5

I, too, thought about using the LSTM trick of trapping the gradient to grow a network to deeper, but didn't start the work just yet. Congratulate the authors for getting there first!

There is possibly a different perspective: a very deep network is a not-so-deep network where layers have sub-layers. Each "layer" has more complex non-linear behavior than a traditional rectifier/sigmoid layer. This is how I personally see inception of GoogLeNet, 3 stacked 3x3 in VGG etc., and I wonder if this is a valid way of looking at highway network as well.

There is one experiment that I would suggest the authors to try, using one thin&tall network to train both MNIST & CIFAR (you can fork at the top for the two domains). What I am interested in is whether, at inference time, the network exhibit the same pattern as found in the networks that are trained separately for each domain, whether the model parameters that are underutilized in MNIST is picked up by CIFAR. In other words, I would love to see how sharing happen in a very deep hierarchy, and whether it is as efficient as possible.

No major complaints. I encourage the authors to try the network on bigger dataset such as ImageNet etc.

Summary: This paper describes an idea inspired by LSTM that allows the training of very deep feedforward network. The idea is simple and elegant, and the analysis are solid.

Author Feedback
Author rebuttal: 